# De novo design of protein minibinder agonists of TLR3

Chloe S. Adams [1,2], Hyojin Kim[3], Abigail E. Burtner [1,2], Dong Sun Lee[3], Craig Dobbins[1,2], Cameron Criswell[1,2], Brian Coventry [1,2,4], Adri Tran-Pearson[1,2], Ho Min Kim [3,5] ✉ & Neil P. King [1,2] ✉

Toll-like Receptor 3 (TLR3) is a pattern recognition receptor that initiates antiviral immune responses upon binding double-stranded RNA (dsRNA). Several nucleic acid-based TLR3 agonists have been explored clinically as vaccine adjuvants in cancer and infectious disease, but present substantial manufacturing and formulation challenges. Here, we use computational protein design to create novel miniproteins that bind to human TLR3 with nanomolar affinities. Cryo-EM structures of two minibinders in complex with TLR3 reveal that they bind the target as designed, although one partially unfolds due to steric competition with a nearby N-linked glycan. Multivalent forms of both minibinders induce NF-κB signaling in TLR3-expressing cell lines, demonstrating that they may have therapeutically relevant biological activity. Our work provides a foundation for the development of specific, stable, and easy-to-formulate protein-based agonists of TLRs and other pattern recognition receptors.

Toll-like receptors (TLRs) are a key family of pattern recognition receptors that act as sentinels of the innate immune system. There are 10 TLRs in humans, each of which recognizes a different pathogen-associated molecular pattern: conserved structural or chemical motifs that distinguish microbial pathogens from self[1]. Structurally, the TLRs are type I transmembrane proteins consisting of an extracellular domain, a single transmembrane helix, and an intracellular Toll/IL-1 receptor (TIR) domain[2,3]. The extracellular domain is responsible for ligand recognition and contains multiple leucine-rich repeats (LRRs) that form a horseshoe-like structure[3–6]. Ligand binding drives TLR dimerization and, in some cases, further association into multimeric complexes[7–12]. Multimerization causes the TIR domain to recruit adapter molecules that activate downstream signaling pathways such as nuclear factor-kappa B (NF-κB) and interferon regulatory factor (IRF) 3 and 7[13,14]. Activation of these signaling pathways leads to the production of pro-inflammatory cytokines, chemokines, and Type I interferons[15–17]. The precise nature of the innate immune response profoundly influences subsequent adaptive immunity[18,19], making targeted modulation of innate immune pathways a promising opportunity to improve the performance of vaccines.

The key role of TLRs in activating the innate immune system makes them ideal targets for vaccine adjuvants: immunostimulators added to vaccines to improve the magnitude, quality, and duration of the adaptive immune response[20,21]. Although aluminum salts are historically the most commonly used adjuvant, over the last quarter century several newer adjuvants with superior performance have been developed that, in some cases, focus on stimulating defined innate immune pathways[22]. For example, a detoxified derivative of the TLR4 agonist lipopolysaccharide (LPS), monophosphoryl lipid A (MPLA), has been formulated with liposomes or aluminum salts in several licensed vaccines[23,24]. The TLR9 agonist CpG has also been licensed as a vaccine adjuvant[25]. Despite this progress, adjuvants are often a bottleneck in vaccine development programs due to intellectual property and safety considerations or formulation difficulties[26,27].

Among the TLRs, TLR3 has several unique features that make it an intriguing target for novel adjuvants. TLR3 is expressed in the

[1]Institute for Protein Design, University of Washington, Seattle, WA 98195, USA. [2]Department of Biochemistry, University of Washington, Seattle, WA 98195, USA. [3]Center for Biomolecular & Cellular Structure, Institute for Basic Science (IBS), Daejeon 34126, South Korea. [4]Howard Hughes Medical Institute, University of Washington, Seattle, WA, USA. [5]Department of Biological Sciences, Korea Advanced Institute of Science and Technology (KAIST), Daejeon 34141, South Korea. ✉e-mail: hm_kim@kaist.ac.kr; neilking@uw.edu

endosome in a variety of cells, including myeloid dendritic cells, macrophages, NK cells, and epithelial cells[28–31]. It is a strong inducer of Type I interferon and is the only TLR that is MyD88-independent, initiating downstream signaling through the TRIF adapter protein[32,33]. Additionally, TLR3 plays an important role in cross-priming[34,35]. Upon ligand binding, TLR3 forms highly organized and cooperative complexes of dimers that cooperatively assemble along linear dsRNA[11]. However, dsRNA is not an ideal adjuvant due to its instability, structural heterogeneity, and promiscuity[16,36]. Stabilized formulations exist (e.g., poly-ICLC) but are still heterogeneous and activate other immune receptors (MDA-5, RIG-1)[37–40]. This promiscuity can lead to over-stimulation and autoimmunity[41]. A chemically well-defined, stable, easy-to-formulate TLR3-specific agonist could alleviate these issues and enable precise, specific induction of TLR3 signaling.

Advances in computational protein design over the last decade have made possible the design of minibinders: small, hyperstable proteins that can bind specifically to target proteins[42–44]. In several cases, minibinders or mimetics of naturally occurring proteins (e.g., cytokines) were shown to be capable of agonizing target receptors and inducing functional biological responses[42,45]. These developments motivate the design of de novo TLR agonists, which could potentially form a novel class of protein-based adjuvants. However, TLRs are notoriously difficult targets: they are highly glycosylated, in many cases have no known protein ligands, and are difficult to express and characterize recombinantly[46,47].

Here, we used computational design to generate de novo protein minibinders of TLR3. We found that multivalent versions of these minibinders cluster the receptor to initiate signaling, suggesting a route to the de novo design of protein-based adjuvants with tailored structural and functional properties.

## Results

### Computational design of TLR3 minibinders

In de novo minibinder design, target site selection significantly influences the success rate of obtaining functional binders[42]. TLR3, like other TLRs, is highly glycosylated, particularly on the surfaces of the molecule not involved in dsRNA-mediated dimerization[5]. Moreover, the carbohydrate-free surface of the molecule features many polar residues with few solvent-exposed hydrophobic patches. The combination of these two features in TLR3 made target site selection difficult, as existing minibinder design methods are most successful when targeting hydrophobic surface patches[42]. Furthermore, since our goal was to design minibinder agonists that can drive TLR3 signaling by multimerizing the receptor, we required a target site that would allow oligomeric versions of the minibinder to simultaneously engage multiple TLR3 ectodomains. Taking these three criteria into account, we selected two target sites on human TLR3 (hTLR3; PDB ID: 1ZIW; ref. 6): Site A, located on the concave face at LRR 19-22 and featuring residues Ile510, Ile534, Ile566, and Ile590; and Site B, spanning LRR 9-11 on the convex face and including residues Leu243, Leu269, Trp273, and Trp296 (Fig. 1a). Although both sites are near N-linked glycans, they nevertheless appeared amenable to minibinder design.

We used the Rosetta RifDock pipeline[42] to design a library of candidate TLR3 minibinders (Fig. 1b). We first used RifGen[48] to dock disembodied amino acid side chains against Sites A and B, and in parallel docked 21,402 pre-existing protein backbones comprising a diverse array of 3-helix bundles against the same sites using PatchDock[43,48]. Docks that matched the most high-scoring disembodied sidechains were output from RifDock and passed to Rosetta FastDesign[43] to generate designed amino acid sequences. The designs were evaluated for predicted binding energy (ddG), contact molecular surface to hydrophobic residues, and spatial aggregation propensity (SAP)[49], and the best-scoring interfaces were extracted using Motif-Graft and subjected to a second round of FastDesign. After filtering using the computational metrics described above, we selected 23,789 designs for experimental characterization: 52% targeting Site B and 48% targeting Site A.

We cloned synthetic oligonucleotides encoding the designs into an expression vector to enable screening by yeast surface display[50] and used fluorescence-activated cell sorting (FACS) to identify cells expressing miniproteins that bound fluorescently labeled TLR3 ecto-domain. After two rounds of sorting with 1 µM streptavidin–tetramerized TLR3 as a probe, we observed a clear population of double-positive cells (Fig. 1c). Enrichment of this population by subsequent sorts without avidity (i.e., with monomeric TLR3) resulted in high numbers of TLR3-binding cells at 150 nM receptor that decreased to near-background levels at 100 nM. Sequencing double-positive cells from each sort identified 11 hits out of

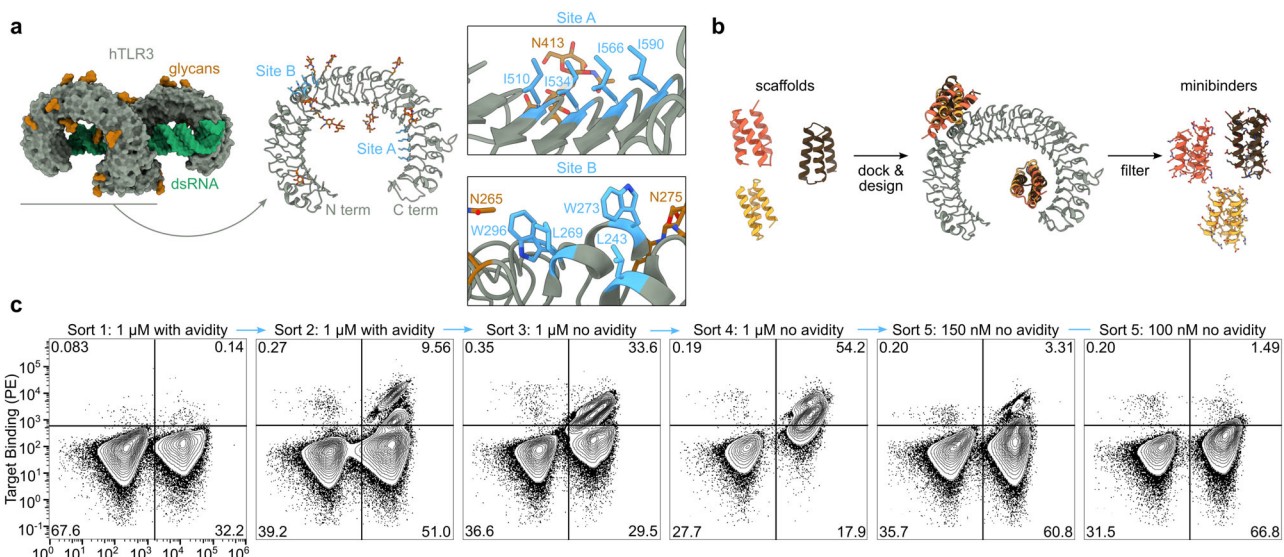

**Fig. 1 | Computational design of TLR3 minibinders. a** Left, TLR3 is natively dimerized and activated by dsRNA (PDB ID: 7WV5). Right, two hydrophobic patches on the TLR3 monomer were targeted for minibinder design (PDB ID: 1ZIW). **b** Polyvaline scaffolds were used in the RifDock pipeline to design de novo miniprotein binders. Several structural metrics were used as filters to select 23,789 designs for experimental screening. **c** Binders were identified using yeast surface display. During Sort 5, binding was clearly observed at 150 nM receptor but approached levels of background signal at 100 nM.

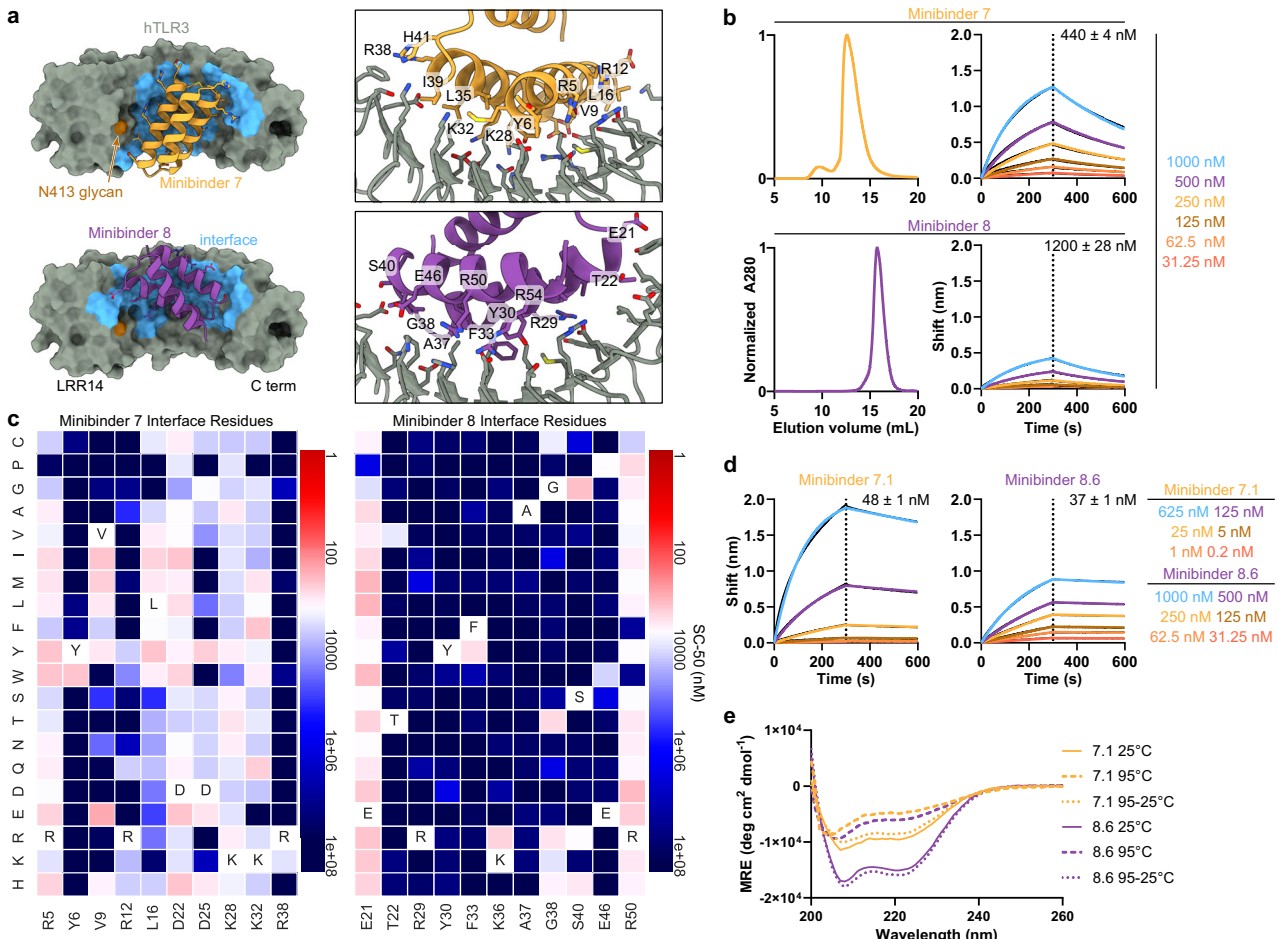

**Fig. 2 | Biochemical characterization and affinity maturation of lead TLR3 minibinders. a** Left, Design models of minibinders 7 (yellow) and 8 (purple) in complex with TLR3 (gray). Right, the details of the predicted interface are shown. **b** Left, Size exclusion chromatograms of each minibinder on a Superdex 75 Increase 10/300 GL. Right, Affinity determination for each minibinder by BLI. The concentrations of hTLR3 used are listed. The black lines represent experimental data, and the colored lines represent fits. KD values are given. **c** Site saturation mutagenesis heat maps of interface residues. The originally designed amino acid at each position is provided at the bottom and in the white square. Red indicates affinity improvement and blue indicates affinity reduction. SC-50 is the midpoint concentration in the binding transition, a proxy for the KD, and is described in detail in ref. 42. **d** Bio-layer interferometry of affinity-matured minibinders 7.1 and 8.6. KD values are given. **e** CD of affinity-matured constructs at various temperatures. Solid line, 25 °C; dashed line, 95 °C; dotted line, 95 °C followed by 25 °C.

23,789 candidates, a success rate consistent with previous applications of this methodology[42].

## Biochemical characterization of TLR3 minibinders

All 11 of the hits identified by FACS target Site A. We suspect that the proximity of Site B to the aforementioned N-linked glycans likely prevented binding at this site. The computational design models of the hits show that they adopt a variety of binding modes within Site A (Fig. 2a, Supplementary Fig. 1). We expressed each with a C-terminal Avi-His$_6$ tag in *E. coli* and purified them by immobilized metal affinity chromatography (IMAC) and size exclusion chromatography (SEC) (Fig. 2b, Supplementary Fig. 1). To validate target binding, the minibinders were biotinylated and their affinities to hTLR3 were determined using bio-layer interferometry (BLI). Six of the eleven minibinders bound to hTLR3 with affinities ranging from 43–1500 nM; minibinders 6–8 yielded the highest binding amplitudes (Fig. 2b, Supplementary Fig. 1).

To evaluate the binding mode of each minibinder and identify mutations that enhance affinity to TLR3, we created a site-saturation mutagenesis (SSM) library for each minibinder by systematically mutating each residue to all 19 other canonical amino acids. We sorted a single, pooled SSM library comprising 11,823 variants based on all 11

original minibinders using FACS and detected binding down to 125 nM, similar to the original library (Supplementary Fig. 2a). Deep sequencing of the sorted libraries allowed us to visualize the effect of each mutation using heatmaps, where red signifies stronger binding and blue indicates weaker binding (Fig. 2c, Supplementary File 1). Minibinders 9 and 10 were deprioritized at this stage due to inconsistencies between their design models and SSM data that suggested these minibinders did not fold as predicted. In other cases the SSM data were consistent with the design models, as indicated by, for example, generally high conservation of amino acids predicted to make substantial interactions with the TLR3. In general, each minibinder had one helix which did not make contact with TLR3 and was less conserved. At some positions, substitutions that appeared to enhance binding to TLR3 could be identified. The interface of minibinder 7 appeared more amenable to a wider variety of interface mutations than minibinder 8, such as L16V/I/M/Y/W in minibinder 7 compared to F33Y in minibinder 8. We selected several such mutations for minibinders 1-8 and 11 and integrated them into combinatorial yeast display libraries for each minibinder using degenerate codons (Supplementary Table 1). During FACS of a pool of these combinatorial libraries, we observed binding at TLR3 concentrations as low as 50 nM, an improvement over the SSM libraries that suggested that combining mutations improved binding

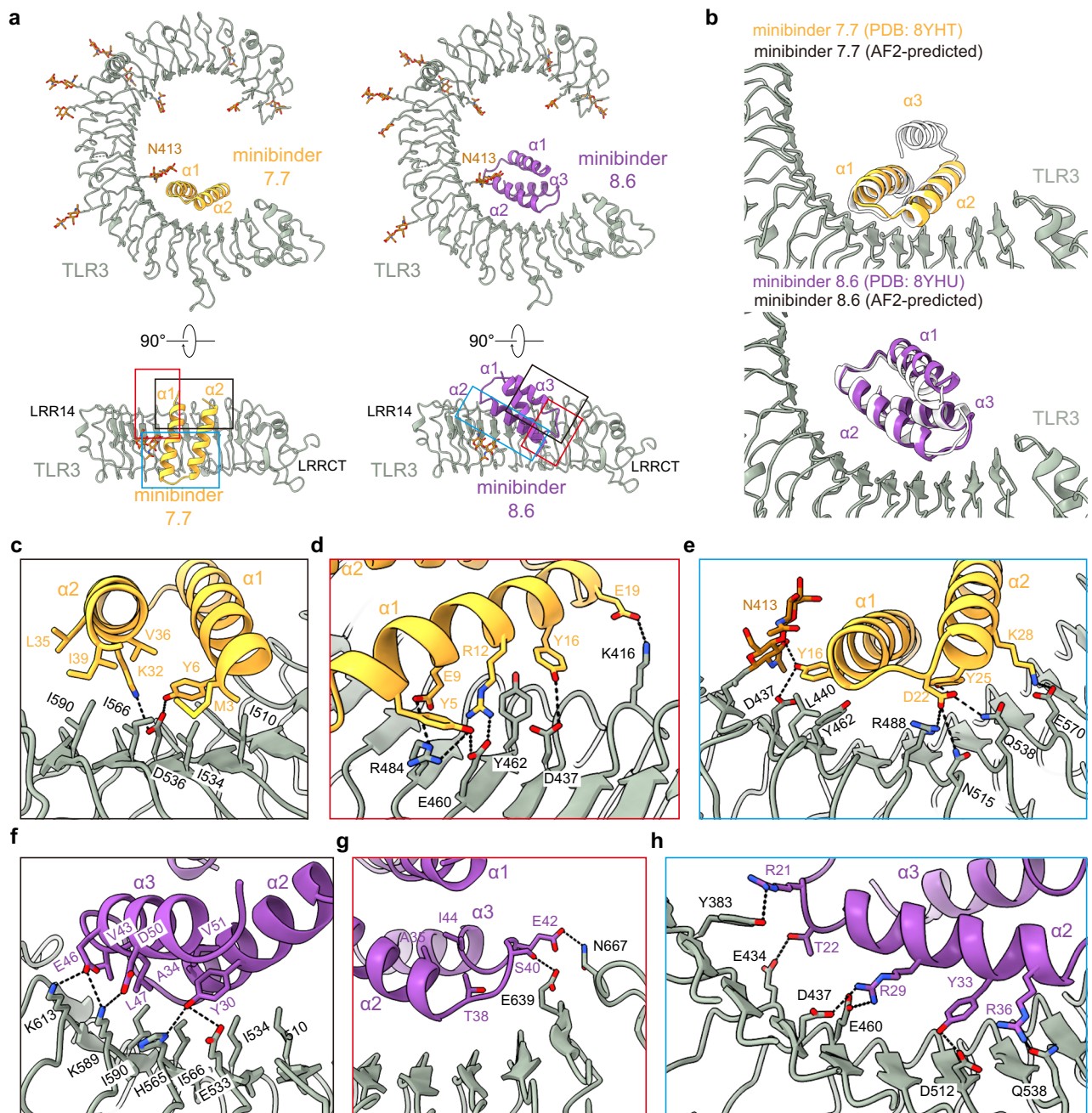

**Fig. 3 | Structural characterization of TLR3 minibinders. a** Two different views of the cryo-EM structures of human TLR3 in complex with minibinder 7.7 (left) or minibinder 8.6 (right). TLR3, glycans, minibinder 7.7, and minibinder 8.6 are in gray, brown, yellow, and purple, respectively. **b** Comparison of experimental and AlphaFold2-predicted structures of the minibinders in complex with TLR3. **c–h** Close-up views of key molecular interactions in TLR3/minibinder 7.7 (**c–e**) or TLR3/minibinder 8.6 (**f–h**). Each box is a close-up view of the same colored box in (**a**). Residues involved in the TLR3/minibinder interaction are displayed as sticks and labeled.

more effectively than single mutations (Supplementary Fig. 2b). All of the hits obtained by deep sequencing in the sorted combinatorial library were derived from minibinder 7. We ordered 12 of these variants, named minibinders 7.1–7.12, for expression in *E. coli* (Supplementary Table 1). We also noted that of the three minibinders with the strongest initial binding amplitudes during BLI (minibinders 6–8), minibinders 6 and 7 had very similar predicted binding modes (Fig. 2a and Supplementary Fig. 1) and eluted at fractions corresponding to higher molecular weights than expected during SEC (Fig. 2b, Supplementary Fig. 1), potentially indicating that they homodimerize. By contrast, minibinder 8 was predicted to bind further from the N413

glycan compared to minibinders 6–7 and eluted from SEC at the expected volume for a monovalent minibinder (Fig. 2a, b). We therefore chose to also optimize minibinder 8 in addition to minibinder 7, and ordered six variants comprising one or more mutations manually selected from the SSM data for expression in *E. coli* (Supplementary Table 1).

We expressed and purified the affinity-matured variants of minibinders 7 and 8 (Supplementary Fig. 3a) and found that each bound hTLR3 with a tighter $K_D$ than the parent minibinder. For minibinder 7 variants, $K_D$ ranged from 1.1–190 nM, whereas for minibinder 8 variants, $K_D$ ranged from 37–690 nM (Fig. 2d; Supplementary Fig. 3a). To

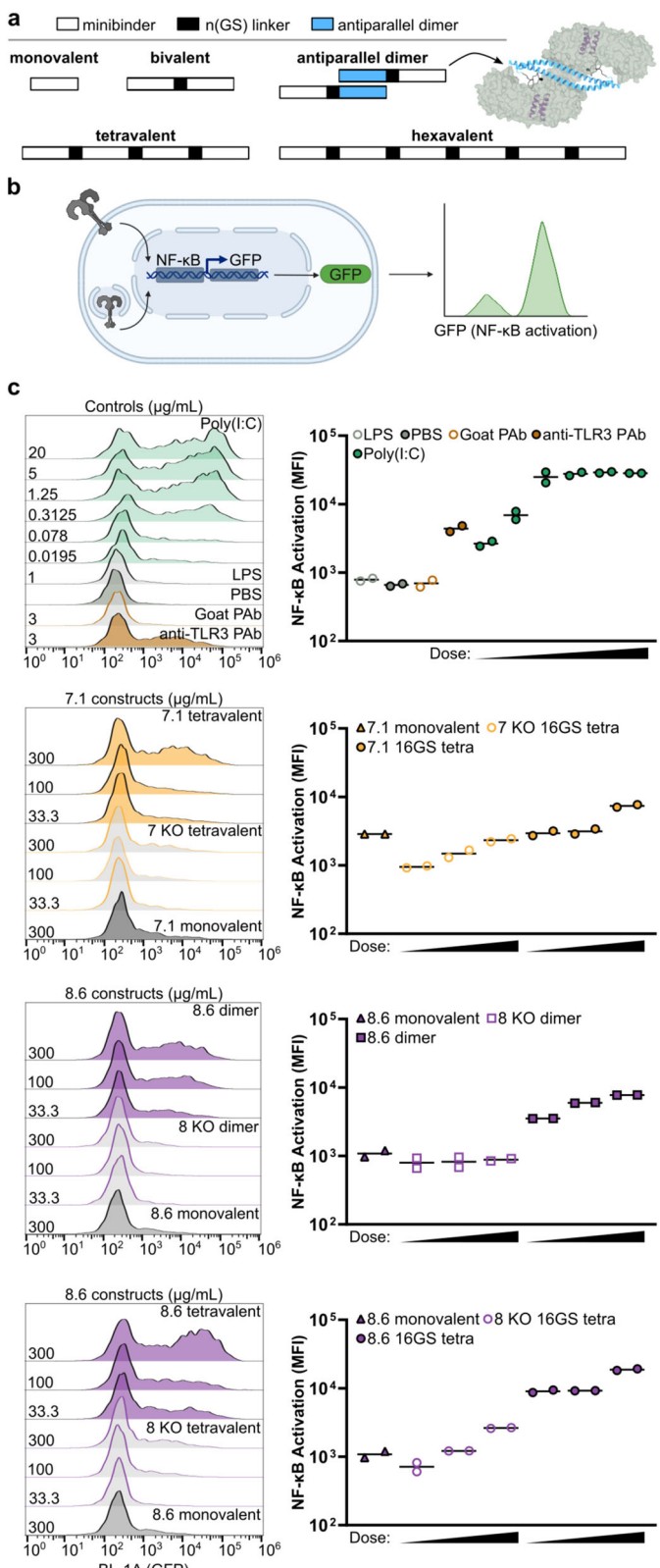

**Fig. 4 | Multimerization of minibinders leads to NF-κB activation. a** Multivalent minibinders were generated by fusing two or more tandem repeats of the minibinder together using 12, 16, or 40-residue (GlySer) linkers. Antiparallel 8.6 dimers were generated by fusing minibinder 8.6 to the C terminus of an antiparallel coiled-coil derived from myosin 10. **b** TLR3 is expressed on the cell surface and in the endosome of TLR3hi cells, which express an NF-κB-linked GFP reporter. GFP levels are measured with flow cytometry. The schematic was rendered using BioRender.com. **c** Left, Histograms showing GFP signal in stimulated TLR3hi cells. The identities and concentrations of the stimuli are provided on the right and left sides of each panel, respectively. Right, mean fluorescence intensity (MFI) values from assay duplicates.

## Structural characterization of minibinder–TLR3 complexes

We selected minibinders 7.7 and 8.6 for structural studies, as these variants exhibited high association rates and low dissociation rates by BLI (Fig. 2d, Supplementary Table 2). We began by mixing minibinder 7.7 with purified human TLR3 (residues K27-A700) in vitro and imaging vitrified specimens by cryo-electron microscopy (cryo-EM). Although a minority of the 2D class averages had density potentially corresponding to the minibinder at target Site A, overall the data suggested low minibinder occupancy. Following a strategy previously established for structural characterization of TLR4/MD-2 complexes[4], we co-expressed the TLR3 ectodomain and each minibinder in HEK293F cells and co-purified the complexes using affinity chromatography followed by SEC. This improved occupancy, and after optimizing grid preparation and data collection to overcome a clear preferred orientation, we determined cryo-EM structures of minibinders 7.7 and 8.6 in complex with TLR3, each at 2.9 Å resolution (Fig. 3, Supplementary Figs. 4 and 5). This resolution permitted us to manually build atomic models of TLR3 and the minibinders into the cryo-EM maps. In both structures, TLR3 closely resembled previously reported structures, exhibiting Cα root mean square deviations (RMSDs) of 1.78 Å for 7WV3 and 1.61 Å for 7C76 (refs. 11,51). Unlike minibinder 8.6, for which clear density was observed for all three α-helices, we could build only α1 and α2 of minibinder 7.7 into well-defined density (Fig. 3a). Interestingly, density corresponding to α3 of minibinder 7.7 could be observed extending from α2 at a low threshold (0.08σ; Supplementary Fig. 6a). Given the proximity of the minibinder to the glycan at N413, we conclude that a steric clash between the glycan and minibinder 7.7 makes α3 protrude out from the complex. This conclusion is supported by our SSM data for minibinder 7—obtained in complex with TLR3—which showed that mutations in the hydrophobic core were more tolerated than would be expected for a well-folded protein (Supplementary File 1). Although it is difficult to discern whether α3 is well-packed against α1 and α2 in the absence of TLR3, we note that CD of the closely related minibinder 7.1 suggested it is less α-helical than minibinder 8.6 (Fig. 2e).

Minibinders 7.7 and 8.6 interacted with the concave surface of TLR3 in slightly different manners, with α1 and α2 of minibinder 7.7 binding to LRR15-LRRCT and α2 and α3 of minibinder 8.6 binding to LRR13-LRRCT. Hydrophobic residues on both minibinders (7.7: M3, Y6, L35, V36, and I39; 8.6: Y30, Y33, A34, V43, L47, and V51) form hydrophobic networks with I510, I534, I566, and I590 in Site A of TLR3 as intended (Fig. 3a, c, f, Supplementary Fig. 6c, d). When the cryo-EM structures were superimposed on the AlphaFold2-predicted models of each complex, the Cα RMSDs between minibinders 7.7 and 8.6 and their predictions were 1.0 and 2.0 Å, respectively (Fig. 3b). For minibinder 8.6, the α2 and α3 helices were well-aligned (Cα RMSD: 1.6 Å), but α1 was slightly displaced from its predicted position (Cα RMSD: 2.7 Å). The altered conformation of minibinder 7.7 α3 described above may result in exposure of hydrophobic residues in the protein core to solvent (e.g., L11 and L30). Alternatively, they may interact with the N413 glycan, although we did not observe ordered density in this region of the map (Supplementary Fig. 6a, c). Conversely, minibinder

obtain initial structural information, we performed circular dichroism (CD) on minibinders 7.1 and 8.6 ($K_D = 48$ and 37 nM, respectively). At 25 °C both minibinders yielded spectra typical of α-helical proteins, although the amplitude of the signal was stronger for minibinder 8.6 (Fig. 2e). Heating the proteins to 95 °C and then returning them to 25 °C showed that both unfolded at high temperature and regained their native structure upon cooling.

8.6 maintained a stable 3-helical bundle through core hydrophobic interactions (Supplementary Fig. 6d).

The interactions between minibinder 7.7 and TLR3 were mainly mediated through electrostatic interactions (Fig. 3c–e and Supplementary Figs. 7 and 8). Specifically, our structural analysis revealed that conserved residues (Y6, R12, E19, and D22) in minibinder 7 and its derivatives engaged in hydrogen bonds or ionic interactions with residues on the concave surface of TLR3. Residues E9, Y25, K28, and K32 of minibinder 7.7 were crucial for forming electrostatic interactions with TLR3 (Fig. 3c–e and Supplementary Figs. 7 and 8). In contrast to minibinder 7.7, the electrostatic interactions mediated by Y33 and conserved residues (R29, Y30, K/R36, E42, E46) in minibinder 8 and its derivatives surround a hydrophobic core interaction network in the minibinder 8.6/TLR3 complex (Fig. 3f, h and Supplementary Figs. 7 and 8). Interestingly, our structural analysis revealed that the interaction of T38 with A35 and I44 stabilizes the loop connecting α2 and α3, facilitating the formation of hydrogen bonds between S40 and E42 of minibinder 8.6 and E639 and N667 of TLR3, respectively (Fig. 3g and Supplementary Figs. 7 and 8). Additionally, the R50D substitution in minibinder 8.6 enabled the formation of an ionic interaction with K589 of TLR3. These observations likely explain why the G38T and R50D substitutions significantly enhanced the binding affinity of minibinder 8.6 compared to 8.4.

### Activation of TLR3 by multivalent minibinders

Like other TLRs, the minimal signaling unit of TLR3 is a ligand-induced dimer[52], although maximal TLR3 signaling requires lateral association of TLR3 dimers, an arrangement facilitated by the linear nature of dsRNA[7,11,12]. We next investigated whether we could engineer our minibinders to drive TLR3 multimerization and thus activation. We approached this problem by generating different multivalent forms of our affinity-matured minibinders. In the first, we genetically fused two or more copies of either minibinder 7.1 or 8.6 in tandem using flexible GS linkers to create bivalent, tetravalent, and hexavalent minibinders that could associate with two or potentially more copies of TLR3 (Fig. 4a). In the second, we genetically fused minibinder 8.6 to the C terminus of an antiparallel dimeric coiled-coil derived from myosin 10 (PDB ID: 2N9B)[53] to generate minibinder dimers that roughly match the geometry of the dsRNA-induced TLR3 dimer (Fig. 4a). We expressed and purified the multivalent 7.1 and 8.6 minibinders as well as the 8.6 antiparallel dimer and confirmed that they bind to TLR3 by BLI (Supplementary Fig. 9a). The multivalent binders appeared to have slower dissociation rates than their monovalent counterparts as expected for avid interactions, although this prevented the calculation of dissociation constants. As negative controls, we also purified interface knockout versions of 7.1 and 8.6 that incorporate mutations shown to decrease TLR3 binding in our earlier SSM data (Supplementary Table 1). We determined by BLI that the interface knockout mutations nearly eliminated TLR3 binding in monovalent and multivalent versions of minibinders 7 and 8 (Supplementary Figs. 3b and 9a). We further assessed the ability of the multivalent constructs to induce TLR3 multimerization via SEC. Both the tetravalent and dimeric forms of 8.6 formed complexes with TLR3 that eluted earlier than monomeric TLR3 (Supplementary Fig. 9b). However, the tetravalent 8.6-TLR3 complex did not elute earlier than the dimeric 8.6 complex, suggesting that the tetravalent 8.6 does not induce the formation of TLR3 assemblies larger than a dimer.

To assess TLR3 activation, we used TLR3hi cells, a previously described HEK293 cell line that expresses TLR3 on the cell surface and features an NF-κB-linked GFP reporter[52] (Fig. 4b). We first confirmed that known TLR3 agonists activated the cells by flow cytometry. Cells stimulated with poly(I:C) at concentrations ranging from 0.3 to 20 µg/mL robustly expressed GFP, and modest GFP expression was still detectable at 0.02 µg/mL (Fig. 4c and Supplementary Fig. 10a).

Polyclonal anti-TLR3 antibody also induced GFP expression as previously shown[52], while a control goat polyclonal antibody did not. Finally, lipopolysaccharide (LPS) failed to induce GFP expression, demonstrating that TLR4 signaling was inactive in the cells.

We stimulated the cells with each of our multivalent minibinders and their corresponding interface knockouts at 300, 100, and 33 µg/mL, as well as the monovalent minibinders at 300 µg/mL. We found that the tetravalent 7.1 and the 8.6 dimer induced comparable levels of dose-dependent GFP expression (Fig. 4c). The tetravalent 7 knockout (KO) yielded substantially reduced but still dose-dependent GFP expression, while GFP expression in cells treated with the 8 KO dimer was not above background. In each case, activation with monovalent minibinder was substantially lower than with the same concentration of multivalent minibinder. The tetravalent 8.6 activated TLR3 more efficiently; GFP expression was higher in cells treated with 33 µg/mL of the tetravalent 8.6 than in cells receiving 9-fold more 8.6 dimer or tetravalent 7.1. At the highest concentration tested, the tetravalent 8.6 induced GFP expression as effectively as moderate concentrations of poly(I:C). Varying GS linker lengths (12, 16, and 40 residues) and minibinder valencies (2, 4, and 6 copies per construct) did not significantly impact TLR3 activation, as all constructs exhibited comparable levels of NF-κB signaling in TLR3-expressing cells (Supplementary Fig. 10b). Although the endosome acidification inhibitor chloroquine had no effect on minibinder-induced signaling (Supplementary Fig. 10c), binding of the minibinders to TLR3 was consistent from pH 5.5–7.4 (in contrast to poly(I:C); Supplementary Fig. 10d), leaving open the possibility that our constructs activate TLR3 either at the cell surface or in the endosome. Finally, although minibinder 7.1 inhibited dsRNA binding to TLR3, especially in its tetravalent form, neither monovalent nor tetravalent minibinder 8.6 inhibited dsRNA binding (Supplementary Fig. 10e). These observations were consistent with our structures of the minibinder–TLR3 complexes, which showed that only minibinder 7.1 would clash with bound dsRNA (Supplementary Fig. 10f). Together, these data establish that our TLR3 minibinders become agonists when multimerized, presumably by clustering multiple copies of TLR3 in close proximity.

## Discussion

Our results demonstrate that computational protein design can be used to create stable protein agonists of TLR3. There are multiple previously solved structures of TLR3 in complex with Fabs or diabodies, all of which target epitopes on the convex surface of the receptor[11,12,54]. Here, minibinders 7 and 8 and their derivatives bind to the concave surface of TLR3. This proved to be a challenging target due to the polar and highly glycosylated surface of the molecule. Due to a nearby glycan, the third helix of minibinder 7.7 was disordered when bound to TLR3, a phenomenon that has not been documented before in designed minibinders to the best of our knowledge. Although this possibly accounts for the weaker agonism of tetravalent 7.1 compared to tetravalent 8.6, it is an intriguing observation that suggests steric competition with glycans or other parts of a target protein structure could be used as a mechanism for intentionally introducing switching behavior in designed minibinders[55].

When multimerized, our minibinders activated NF-κB via TLR3 in a cell line that expresses TLR3 on the cell surface. Although these molecules have the potential to become protein-based adjuvants, further work will be required to evaluate them as such. For example, they must be engineered to reach the endosome of cells expressing the receptor in vivo. Furthermore, previous studies showed that the minimal TLR3 signaling complex in the TLR3hi cell line is a dimer[52]. However, several additional reports, including recent cryo-EM structures of TLR3 in complex with dsRNA, have shown that the formation of linear polymers of TLR3 dimers along longer dsRNA molecules is required for maximal activation[7,11,12]. It is likely that the activity of our minibinder agonists could be improved by multimerizing them so that they match the geometry of the fully active signaling complex.

Supporting this notion, a recent study of designed multimeric cytokines established that tuning the precise geometry of receptor clustering through design can significantly impact activation and signaling[56]. Combining our approach to TLR minibinder design with recently developed generative methods for designing multimeric scaffolds[57] could become a powerful and general approach to designing protein-based TLR agonists.

## Methods

### Computational design

The ectodomain of TLR3 (PDB ID: 1ZIW) was used for the design. The PDB was truncated to save computational time. Minibinders were designed as outlined in ref. [42]. In brief, RifGen[48], PatchDock[58], RifDock[48], FastDesign[43], and the MotifGraft mover[59] were used. Millions of designs were created using this pipeline and then filtered on various Rosetta metrics (contact molecular surface to hydrophobic residues, ddG, and SAP)[42]. Designs before and after the MotifGraft stage were ordered.

### Library preparation

Designs were padded to 65 amino acids with serine. Then, the designs were codon-optimized for expression in *Saccharomyces cerevisiae*. Oligonucleotides encoding the designs and SSM mutations were purchased from Agilent technologies. Libraries were amplified as previously described[42].

### Yeast surface display

Minibinder libraries were transformed into yeast via electroporation. Minibinders were displayed on the surface of yeast with a C-terminal myc tag to enable detection of cell surface expression using a FITC-labeled anti-myc antibody. Biotinylated hTLR3 ectodomain was recognized by streptavidin–PE. An initial sort was done to collect cells that express the designs (FITC+). These cells were subsequently sorted against decreasing concentrations of hTLR3 with and without avidity.

The *Saccharomyces cerevisiae* EBY100 strain was cultured in C-Trp-Ura medium containing 2% (w/v) glucose (CTUG). For induction, cells were harvested by centrifugation at 5000$g$ for 5 min and subsequently resuspended in SGCAA medium supplemented with 0.2% (w/v) glucose at a cell density of $1 \times 10^7$ cells per mL. The cells grew at a temperature of 30 °C for 16–24 h. After induction, the cells were washed using PBSF (PBS with 1% (w/v) BSA). The cells were labeled with anti-c-Myc fluorescein isothiocyanate (FITC, Miltenyi Biotech) and streptavidin-phycoerythrin (SAPE, ThermoFisher).

During the initial sorting of the naive library, an avidity-based approach was employed. This entailed simultaneous incubation of the biotinylated target, SAPE, and FITC. In contrast, sorts performed without avidity involved a sequential procedure. Specifically, cells were incubated with the biotinylated target, washed with PBSF, and then incubated with SAPE and FITC. For SSM libraries, two rounds of no-avidity sorts were performed and in the third round of screening the libraries were titrated with a series of decreasing concentrations of hTLR3. After each sort, cells were collected for deep sequence analysis. For the combinatorial libraries, five rounds of no-avidity sorts were performed with decreasing concentrations of hTLR3. Only the top 0.1% of the binding population was collected after each sort. These populations were plated on C-trp-ura plates and the sequences of individual clones were determined by Sanger sequencing.

### Deep sequence analysis

Collected sorts for the naive and SSM libraries were sequenced using Illumina NextSeq sequencing. The PEAR program[60] was used to assemble the fastq files as previously described[42]. For the SSM libraries, the apparent SC-50 was estimated using the fitting procedure as described in ref. [42]. SC-50 is the midpoint concentration in the binding transition, a proxy for $K_D$. Heatmaps were generated in Jupyter notebooks based on the scripts in ref. [42].

### Combinatorial library preparation

We identified mutations that enhanced affinity from SSM heatmaps. We then used SwiftLib to create libraries containing these mutations with degenerate codons[61]. We ordered two overlapping Ultramers (long ssDNA oligos) for each design that contained the selected degenerate codons (Integrated DNA Technologies), assembled them by PCR, and cloned into yeast. All libraries were combined before yeast surface display.

### Protein expression and purification

Genes encoding minibinders were synthesized and cloned by Integrated DNA Technologies into a pET-29b(+) vector. Plasmids were transformed into BL21(DE3) (NEB) and 30 mL of single-colony culture was grown in Terrific Broth II (MP Biomedical) at 37 °C overnight. In total, 25 mL was transferred to 0.5 L of autoinduction medium and grown at 37 °C for 2 h and 18 °C overnight[62]. Cells were collected by spinning for 30 min at 4000$g$. Pellets were resuspended in lysis buffer (50 mM Tris pH 8.0, 250 mM NaCl, 20 mM imidazole, 0.04 mg/mL RNase, 0.1 mg/mL lysozyme, 0.1 mg/mL DNase, 1 mM PMSF). The cells were lysed via a microfluidizer. The lysed cells were clarified via centrifugation at 14,000$g$ for 30 min. The supernatant was run over Nickel–NTA resin and washed with 3 CV of wash buffer (50 mM Tris pH 8.0, 250 mM NaCl, 20 mM imidazole). Protein was eluted with an elution buffer (50 mM Tris pH 8.0, 250 mM NaCl, 500 mM imidazole) and further purified by size exclusion chromatography using a Superdex 75 Increase 10/300 GL column (Cytiva). The multivalent minibinders were purified using a Superdex 200 Increase 10/300 GL column (Cytiva).

The TLR3 ectodomain used for yeast display and BLI was obtained from Wuxi Biologics (custom order). The sequence is derived from Choe et al.[6] and included a Gp67 signal peptide and C-terminal His and Avi tags (Supplementary Table 1). To produce the protein, a baculovirus was generated using Sf9 cells and High Five cells were used for protein expression. The final buffer was 50 mM Tris, 250 mM NaCl, 1 mM DTT.

### Bio-layer interferometry

BLI was performed on an Octet RED96 or Octet R8. All biosensors were hydrated in kinetics buffer HBS-EP+ (10 mM HEPES, 150 mM NaCl, 3 mM EDTA, 0.05% v/v surfactant P20) with 0.5% w/v non-fat dry milk (Cytiva). Minibinders were biotinylated and excess biotin was purified out by SEC. Biotinylated minibinders were diluted to a final concentration of 0.01 mg/mL in kinetics buffer and loaded onto streptavidin biosensors (Sartorius). TLR3 was diluted in kinetics buffer and its association was measured for 300 s, followed by a dissociation for 300 s in kinetics buffer.

For multivalent constructs, Human TLR3-FC (Acro Biosystems) was diluted to 0.01 mg/mL in kinetics buffer and loaded onto Protein A tips (Sartorius). The multivalent constructs were diluted in kinetics buffer and their association was measured for 300 s, followed by a dissociation for 300 s in kinetics buffer. Constructs were tested at 10, 5, and 1.25 μM. Interface knockout constructs were tested at 10 μM.

### Circular dichroism

CD measurements were carried out on a JASCO J1500 spectrometer at 25–95 °C, using a 1 mm path-length cuvette, at wavelengths from 200 to 260 nm. Proteins were measured at 0.4 mg/mL in PBS buffer.

### Constructs for recombinant expression of TLR3/minibinder complexes

Human TLR3 ectodomain (residues K27-A700) followed by a 6× His-tag was cloned into the BamHI and XbaI sites of the pcDNA 3.1 vector (#V79020, Invitrogen) containing the vascular endothelial growth

factor receptor 1 (VEGFR) signal sequence for protein secretion. Residues S1-L56 of minibinder 7.7 or residues S1–S56 of minibinder 8.6 followed by a thrombin cleavage site and Protein A-tag were cloned into the BamHI and EcoRI sites of the pcDNA 3.1 vector containing the VEGFR signal sequence for protein secretion.

## Expression and purification of the TLR3/minibinder complex for cryo-EM

A total of 1 µg of plasmid DNA (TLR3:minibinder-protein A ratio of 1:4) was transfected into $2.5 \times 10^6$ Expi 293F cells (#A14527, Thermo Fisher Scientific) using Expifectamine (#A14524, Thermo Fisher Scientific), and cells were cultured in Expi293 expression medium (#A14351, Thermo Fisher Scientific) at 37 °C and 8% $CO_2$ with shaking (orbital shaker, 120 rpm) for 4 days. After centrifugation to remove the cells, the culture supernatant was loaded onto IgG Sepharose 6 Fast Flow (#17-0969-01, Cytiva). After washing with 10 column volumes of wash buffer (20 mM Tris-HCl pH 8.0, 200 mM NaCl), the Protein A–fused binary complexes (TLR3/minibinder-protein A) bound to the resin were incubated with thrombin (1% [v/v] in wash buffer) at 4 °C for 16 h to remove the C-terminal Protein A tag. The eluted TLR3/minibinder complexes were concentrated to 1 mg/mL using an Amicon Ultra centrifugal filter (#UFC8030, Millipore) and further purified by SEC using a Superdex 200 Increase 10/300 GL column (Cytiva) equilibrated in wash buffer (20 mM Tris-HCl pH 8.0, 200 mM NaCl). The peak fraction was used for cryo-EM analyses without concentration.

## Cryo-EM sample preparation and data collection

Initially, the use of a general quantifoil grid led to a preferred orientation in the 2D average of the initial dataset, showing only C-shaped side views of the TLR3/minibinder complex. This limitation caused the 3D volume to stretch sideways, preventing the generation of a high-resolution reconstruction. It is known that coating the cryo-EM grid with a clean graphene oxide layer can help the protein particles adopt various orientations[63]. Indeed, graphene oxide-coated grids resulted in 20% of views from other angles, while approximately 80% of the particles still displayed the C-shaped view. Graphene oxide-coated Quantifoil R1.2/1.3 300 mesh copper holey carbon grids (Quantifoil Micro Tools) were glow discharged using a PELCO easiGlow Glow Discharge Cleaning system (Ted Pella) for 5 s at 10 mA. A 3 µL of the purified TLR3/minibinder complexes were applied to the grid and incubated for 10 s in 100% humidity at 4 °C. After 2 s of blotting, the grid was plunged into liquid ethane using a Vitrobot MkIV (Thermo Fisher Scientific). Micrographs were acquired on a Titan Krios G4 TEM operated at 300 keV with a K3 direct electron detector (Gatan) at the Institute for Basic Science (IBS), using a slit width of 20 eV on a GIF-quantum energy filter. EPU software was used for automated data collection at a calibrated magnification of 130,000× under the single-electron counting mode and correlated-double sampling (CDS) mode[64], yielding a pixel size of 0.664 Å/pixel. Detailed image acquisition parameters for each TLR3/minibinder complex are summarized in Supplementary Table 3.

## Image processing, model building, and refinement

The detailed image processing workflow and statistics are summarized in Supplementary Figs. 4 and 5 and Supplementary Table 3. Raw movies were motion-corrected using MotionCorr2[65], and the CTF parameters were estimated by CTFFIND4[66]. All other image processing was performed using cryoSPARC v.4.2.1[67]. Initially, particles were picked with a manual picker of cryoSPARC from a few micrographs. 2D class averages representing projections in different orientations selected from the initial 2D classification were used as templates for automatic particle picking from whole micrographs. The resulting particles went through subsequent 2D classifications for cleanup within cryoSPARC.

Even though the preferred orientation was slightly improved using the graphene oxide-coated grid, approximately 80% of the particles were still included in the C-shaped views. To find the best ratio between views that can create a nice 3D volume by making the Euler angle distribution as even as possible, we tested various ratios of C-shaped views and other views using Rebalance 2D in cryoSPARC. Robust 3D volumes were generated using twice as many C-shaped particle views as other views by removing ~40% of the C-shaped view from the dataset. Non-uniform refinement[68], local motion correction[69], and CTF refinement[70] improved the particle alignment and map quality. The final refinement yielded a map at an overall ~3.0 Å resolution (with tight mask). The mask-corrected Fourier shell correlation (FSC) curves were calculated in cryoSPARC, and reported resolutions were based on the gold-standard Fourier shell correlation (FSC) = 0.143 criterion[71]. Local resolutions of density maps were estimated by Blocres[72].

Model building for TLR3 was initiated by docking the AlphaFold2-predicted TLR3 into the post-processed cryo-EM map generated from DeepEMhancer[73] using the Phenix package. The Cα chain and side chains of minibinders were manually built in the cryo-EM density map using Coot[74], referring to their AlphaFold2-predicted structure. Models of the TLR3/minibinder complexes were manually adjusted in Coot and refined against the map by using the real space refinement in the Phenix package. The refinement statistics from Phenix validation are summarized in Supplementary Table 3.

## Cell assay

HEK293-TLR3hi cells were cultured in DMEM with 4 mM L-glutamine (ThermoFisher), 10% FBS (ThermoFisher), 1% penicillin–streptomycin (ThermoFisher), and 500 µg/mL G418 sulfate (ThermoFisher). In total, $10^5$ cells were plated in a 24-well plate in 0.25 mL of media for 3 h. Ligand was added and 24 h later NF-κB-driven GFP expression was measured using flow cytometry on an Attune NxT. Poly(I:C) HMW (InvivoGen) and polyclonal goat anti-TLR3 (R&D systems) were used as positive controls. LPS-B5 (InvivoGen) and polyclonal goat IgG (R&D systems) were used as negative controls. When used, chloroquine (InvivoGen) was added to cells 1 h prior to stimulation.

## SEC purification assay

8.6 16GS tetravalent and 8.6 dimer were incubated with hTLR3 ectodomain (8.6 16GS tetravalent:hTLR3 molar ratio of 30:1; 8.6 dimer:hTLR3 molar ratio of 24:1) for 12 h at 4 °C and then purified by size-exclusion chromatography using a Superdex 200 Increase 10/300 GL column (Cytiva) equilibrated with 20 mM Tris pH 8.0 and 200 mM NaCl.

## pH binding ELISAs

Biotinylated minibinder was diluted in PBS (pH 5.5, 6.5, 7.4) to 50 µg/mL and Poly(I:C)(HMW) biotin (InvivoGen) was diluted in PBS (pH 5.5, 6.5, 7.4) to 2.5 µg/mL. For each construct, 100 µL was plated onto Pierce™ Streptavidin Coated High Capacity Plates (Thermo-Fisher). Plates were incubated for 1 h at 25 °C. Plates were blocked with 200 µL of blocking buffer (PBS + 5% non-fat milk) (pH 5.5, 6.5, 7.4) for 1 h at 25 °C. Plates were washed 3× in PBST (PBS + 0.1% (v/v) Tween-20) (pH 5.5, 6.5, 7.4) using a plate washer (BioTek). Then, 100 µL of hTLR3-FC (Acro Biosystems) (pH 5.5, 6.5, 7.4) was added to each well and incubated for 1 h at 25 °C. Plates were washed 3× in PBST, then anti-human IgG Fc HRP conjugated (#ab97225, abcam) was diluted 1:10,000 and 100 µL was added to each well and incubated at 25 °C for 30 min. Plates were washed 3× in PBST and 100 µL of TMB (SeraCare) was added to every well for 2 min at room temperature. The reaction was quenched with the addition of 100 µL of 1 N HCl. Plates were immediately read at 450 nm on an Epoch2 plate reader.

## Inhibition ELISAs

Pierce™ Streptavidin Coated High Capacity Plates were coated with 1.5 µg/mL of Poly(I:C)(HMW) biotin at pH 6.5. Plates were incubated for 1 h at 25 °C. Plates were blocked with 200 µL of blocking buffer (PBS + 5% non-fat milk) (pH 6.5) for 1 h at 25 °C. Plates were washed 3× in PBST (pH 6.5). In total, 20 µg/mL hTLR3-FC was incubated 1:1 with a dilution series of minibinders in blocking buffer for 10 min. In total, 100 µL of the complexes were added to the plate and incubated for 1 h at 25 °C. Plates were washed 3× in PBST, then anti-human IgG Fc HRP conjugated (#ab97225, abcam) was diluted 1:10,000 and 100 µL was added to each well and incubated at 25 °C for 30 min. Plates were washed 3× in PBST and 100 µL of TMB (SeraCare) was added to every well for 2 min at room temperature. The reaction was quenched with the addition of 100 µL of 1 N HCl. Plates were immediately read at 450 nm on an Epoch2 plate reader.

## Reporting summary

Further information on research design is available in the Nature Portfolio Reporting Summary linked to this article.

## Data availability

The cryo-EM density maps of TLR3/minibinder 7.7 and TLR3/minibinder 8.6 complex have been deposited in the Electron Microscopy Data Bank (EMDB; https://www.ebi.ac.uk/pdbe/emdb/) under accession number EMD-39300 and EMD-39301, respectively. The atomic coordinates of TLR3/minibinder 7.7 and TLR3/minibinder 8.6 complex have been deposited in the Protein Data Bank (PDB; https://www.rcsb.org) with accession code 8YHT and 8YHU, respectively. The structural data from PDB used in this study are listed below: 7WV5 (TLR3 ectodomain/poly(I:C) complex), 1ZIW (hTLR3 ectodomain), 7WV3 (TLR3 linear cluster), 7C76 (hTLR3/UNC93B1 complex), 2N9B (antiparallel dimer). All other data are available in the manuscript or the supplementary materials. Further information and requests for resources and reagents should be directed to and will be fulfilled by the corresponding authors (hm_kim@kaist.ac.kr) (neilking@uw.edu). Source data are provided with this paper.

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

## Acknowledgements

We would like to thank Andrew Borst, Jung Ho Chun, Xinru Wang, Joseph Harman, Jing Yang Wang, Naveen Jasti, Naveen Mehta, Daniel Humphrys, Madison Kennedy and Sidney Lisanza for helpful discussions; Joshua Leonard and Amparo Cosio for generously providing TLR3hi cells; and Inna Goreshnik, Aza Allen, Cami Cordray, Samer Halabiya, and Dionne Vafeados for help with yeast library preparation. We are grateful to the staff of the Research Solution Center at IBS for help with cryo-EM data collection. Cryo-EM data processing was performed on Olaf, the data analysis hub in the IBS Research Solution Center. This work was supported by grants from the Bill & Melinda Gates Foundation (INV-010680 and INV-043758 to N.P.K.), the Institute for Basic Science (IBS-R030-C1 to H.K. and H.M.K.), the NIH (T32GM008268 to C.S.A.). A.B. was supported by a Washington Research Foundation Fellowship, the Barry Goldwater Scholarship, and the Audacious Project at the Institute for Protein Design. Figure 4b was created in BioRender (BioRender.com/y04e628).

## Author contributions

C.S.A. and B.C. designed minibinders. C.S.A. and C.C. performed computational analysis of designs. C.S.A. and A.E.B. screened designs and carried out biochemical characterization. C.S.A., A.T., and C.D. performed cell-based assays. H.K., D.S.L., and H.M.K. designed and carried out electron microscopy experiments. N.P.K. and H.M.K. supervised research. C.S.A., H.K., H.M.K., and N.P.K. wrote the paper.

## Competing interests

A provisional patent application has been filed by the University of Washington on the TLR3 minibinders described here, listing C.S.A., B.C., and N.P.K. as co-inventors. The King lab has received unrelated sponsored research agreements from Pfizer and GSK. The remaining authors declare no competing interests.
