## [Transparent Peer Review file · Nature Communications]

De novo design of protein minibinder agonists of TLR3

Corresponding Author: Professor Neil King

Version 0:

Reviewer comments:

Reviewer #1

(Remarks to the Author)

The manuscript by Adams et al. describes de novo design of protein binders targeting TLR3 with high affinity. The authors also show the CryoEM structures of these binders bound to TLR3, and the multimerized binders and the binders fused with coiled-coil domain can activate TLR3. The binding site of the binder is completely different from dsRNA, the natural ligand of TLR3. This work demonstrates that computational protein design successfully creates the binder which can act as an agonist. This idea in this work can be applicable to other TLRs for developing agonist and/or antagonist. The paper is interesting but the authors should address the concerns.

1) There is no doubt that the mini-binder(s) bind to the concave surface of TLR3, confirmed by biophysical analysis and the direct structure determination by CryoEM analysis. On the other hand, I wonder how the multimerized binder and coiled-coil fused mini-binder bind to and activate TLR3. My concerns are as follows:

Firstly, what is the stoichiometry between TLR3 and these binders (the multimerized binder and coiled-coil fused mini-binder)? Probably, the stoichiometry between TLR3 and the coiled-coil fused mini-binder might be 2:2, but how about the multimerized binder? Is it possible to estimate the stoichiometry by SEC or other appropriate analysis?

Secondly, the activated form of TLR3 bound to the binders (the multimerized binder and coiled-coil fused mini-binder) is similar to TLR3-dsRNA structure? Ideally, authors determine the 3D structure by CryoEM, but 2D class imaging by CryoEM should be shown.

Thirdly, the repeat number of GS linker (i.e. length of GS linker) affect the activity?

2) As the authors addressed, the mini-binders activated NF- κ B via TLR3 in a cell line that expresses TLR3 on the cell surface. TLR3 is localized in endolysosome, therefore mini-binders should bind to TLR3 at an acidic condition. Binding affinity of the mini-binders to TLR3 should be shown at an acidic condition.

3) The mini-binders do not inhibit dsRNA binding to TLR3?

Reviewer #2

(Remarks to the Author)

The purpose of the study is to synthesise and test peptide agonists and antagonists of the innate immune receptor TLR3. This is a challenging objective because TLR3 is naturally activated by dsRNA localised to endosomes and involves dimerization and multimerization of the receptor. Peptides will not usually be efficiently taken up by the endosomal pathway so the study relies on the use of a HEK293 cell line in which TLR3 can signal from the cell surface.

The TLR3 ectodomain is a challenging target as it is highly glycosylated so the authors identified two sites on the concave surface of the solenoid relatively free of glycans. Initial in silico docking analysis identified candidate peptides and these were then screened by yeast surface display yielding 11 hits. Further affinity maturation yielded two 'minbinders' with nM affinities for site A (7.1 and 8.6) and subsequent structural studies revealed similar α -helical bundles bound at site A. This peptide design is impressive and a proof of principle for targeting difficult protein interfaces.

The next step was to test the ability of the minibinders to affect TLR3 signalling. To do this they made a dimeric and tetrameric forms of the binders and tested for activation in HEK293 (TLR3 high) cells. They chose to use an NF κ B reporter linked to GFP as a readout and these results are presented in Figure 4. I do not understand why they have not use more

usual and more quantifiable outputs such as luciferase or SEAP. In Figure 4d what is BL-1A and in what way is this a histogram? The concentration of minibinders needed to produce a response is very high (300µg/ml). I find Fig4e, which is key to the whole study, difficult to understand. The symbols used for the different experiments are unclear – why does 7KO tetramer still signal? Why are there two 8.6 dimers one signalling and the other not (purple lined squares and purple filled squares)? What is the yellow filled triangle at the left half way up the y-axis? In my opinion these experiments should be repeated with a luciferase reporter in a more relevant cell line eg THP1-Dual-hTLR3 cells. They should also be supplemented with TRIF cytokine assays like IFN and RANTES. The cell surface origin of the signal should be validated with an endocytosis blocker. Perhaps also a control using ordinary HEK293s should be added – in our experience these do signal in response to polyIC but at a lower level.

In conclusion the development of peptides agonists targeted at the TLR3 is an important result but the signalling properties of the helical peptides needs further characterization.

Reviewer #3

(Remarks to the Author)

This interesting and very well written manuscript presents the de novo design of TLR3 binding miniproteins, and the assembly of these binders into multivalent constructs with agonist activity. However, no novel protein design or engineering advances are reported. Therefore, the novelty seems to come primarily from the application.

Unfortunately, the functional characterization of the engineered constructs to act as an adjuvant does not go further than the initial proof-of-concept for TLR3 signaling using an artificial reporter cell line. Indeed, the minibinder was not evaluated for activity as an adjuvant, presumably because a new construct would need to be engineered that binds to TLR3 from a model organism like a mouse or rat? Nor was the binding affinity, valency, or linker distance systematically varied and evaluated in order to tune the agonist activity to achieve the desired effect as an adjuvant.

Nonetheless, the protein design work is well performed and the biochemical and structural characterization of the designs is thorough. Very few academic labs in the world are able to engineer a minibinder in the way that the authors describe, so the application of this technology remains exciting.

The authors position this as a first step with functional work to come in the future. However, without compelling functional data, this story should not focus so thoroughly on the use of these constructs as an adjuvant, and should focus instead more on the protein design, biochemistry, and structural biology aspects, as that is where the strength of the data lies. Pending this repositioning of the story to align the narrative with the data reported, I highly recommend this manuscript for publication in Nature Communications.

Additionally, I have the following comments -

On page 3, the sentence “We first used RifGen 48 to dock disembodied amino acid side chains against Sites A and B, and in parallel docked 21,402 pre-existing polyvaline scaffolds (i.e., backbones) against the same sites using PatchDock.” The terms “scaffold” and “backbone” are both protein design jargon and using one does not necessarily help the uninitiated reader understand the other. I think it might be helpful to explain these terms in plain language, i.e. a “scaffold” or “backbone” is a protein main chain or tertiary structure.

Authors mention that “TLR3 is a challenging target for several reasons” but don’t describe any new computational innovations that were required to get binders for this target. If previously developed workflows could still be employed and binders were found at a hit rate that is on par with previous studies, is it fair to say that this is a challenging target?

For the cryo-EM structure determination, I’m a bit confused as to why mixing minibinder with the TLR3 ectodomain failed to produce sufficient occupancy, but co-expression was. It would be helpful to include a bit of discussion to explain what you think is happening here.

Fusing multiple minibinders together is a great strategy, but the nomenclature used here is confusing because it conflicts with the broader protein biochemistry literature. “Tetramer” is a term used to describe 4 separate protein chains that bind each other to form a quaternary structure complex. This term is not commonly used to describe genetic fusions of a protein into a single construct, and I worry the way that it is used here will be confusing to readers. For example, an IgG heavy chain consists of 4 separate Ig domains in a single protein construct, but this is never referred to as a tetramer of Ig domains in the literature; it’s one protein chain. Perhaps “multivalent”, “tetra-valent” and “bi-valent” are better terms to use here? Whatever the authors choose, the term should not have a different meaning that is already well established in the literature.

It looks like the BLI for the constructs named ‘8.6 dimer’ and ‘8.6 tetramer’ have a slower koff, which is what you would expect from getting an avidity boost. It might be helpful to include a sentence describing as much, and perhaps to mention the difference in koff that you measured for the monovalent and multivalent constructs. Also, what is the final KD that you measured for the multi-valent constructs?

The minibinder 7.7 needs to partially unfold to bind. How stable is this protein to serum proteases? Would you expect it to

survive in vivo long enough to be an effective adjuvant? Would a strong immune reaction against these minibinders neutralize their activity and prevent them from functioning as adjuvants?

The authors never define what "SC-50" means, e.g. Figure 2D. Furthermore, the methods section does not contain any information for how the SSM library was sorted, how the data were analyzed to produce the heatmaps, or what the measurements represent.

In the Combinatorial library preparation section what does it mean that "ultramers were stitched together and cloned into yeast"? Does this mean that DNA was assembled enzymatically in vitro prior to yeast transformation? If so, how? How were the yeast transformed, i.e. chemical competence or electroporation? What fold-coverage was obtained for the library?

Version 1:

Reviewer comments:

Reviewer #1

(Remarks to the Author)

The authors have addressed my previous concerns. I recommend the publication of this manuscript in Nature communications.

Reviewer #2

(Remarks to the Author)

On the whole the authors have responded adequately to my comments, especially they have tidied up Figure 4 and it is now more comprehensible. However they have side-stepped two of my points: firstly they have done some experiments with chloroquine to satisfy the concern about pH dependence (lack of) and this looks convincing. It should be noted that chloroquine has lots of other off-target effects and the more specific bafilomycin should have been used. They refer to this same data as a response to my point about endocytosis but chloroquine is not an endocytosis inhibitor, they should use dynasore. Secondly, they claim that validation THP-1 macrophages would be confounded by LPS contamination. I do not accept that - it would be straightforward to inhibit TLR4 with the very specific TAK-242 as a control for LPS.

Reviewer #3

(Remarks to the Author)

The authors have address all my questions and comments, and I now recommend this manuscript for publication in Nature Communications.

Version 2:

Reviewer comments:

Reviewer #2

(Remarks to the Author)

The two substantive points I made have now been addressed with experimentation and I am happy to recommend for Nat Comm.

Reviewer #1:

The manuscript by Adams et al. describes de novo design of protein binders targeting TLR3 with high affinity. The authors also show the CryoEM structures of these binders bound to TLR3, and the multimerized binders and the binders fused with coiled-coil domain can activate TLR3. The binding site of the binder is completely different from dsRNA, the natural ligand of TLR3. This work demonstrates that computational protein design successfully creates the binder which can act as an agonist. This idea in this work can be applicable to other TLRs for developing agonist and/or antagonist.

We thank the reviewer for their positive assessment of our work.

The paper is interesting but the authors should address the concerns.

1) There is no doubt that the mini-binder(s) bind to the concave surface of TLR3, confirmed by biophysical analysis and the direct structure determination by CryoEM analysis. On the other hand, I wonder how the multimerized binder and coiled-coil fused mini-binder bind to and activate TLR3. My concerns are as follows:

Firstly, what is the stoichiometry between TLR3 and these binders (the multimerized binder and coiled-coil fused mini-binder)? Probably, the stoichiometry between TLR3 and the coiled-coil fused mini-binder might be 2:2, but how about the multimerized binder? Is it possible to estimate the stoichiometry by SEC or other appropriate analysis?

We thank the reviewer for this question. Based on the SEC data shown in an updated Extended Data Fig. 9b, tetravalent 8.6 and dimeric 8.6 form complexes with TLR3 that elute earlier than monomeric TLR3, consistent with complex formation. However, the tetravalent 8.6/TLR3 complex does not elute earlier than the dimeric 8.6 complex, suggesting that the tetravalent 8.6 does not induce the formation of TLR3 assemblies larger than a dimer. We have added the following text to the Results section to describe these data:

“We further assessed the ability of the multivalent constructs to induce TLR3 multimerization via SEC. Both tetravalent and dimeric forms of 8.6 formed complexes with TLR3 that eluted earlier than monomeric TLR3 (**Extended Data Fig. 9b**). However, the tetravalent 8.6/TLR3 complex did not elute earlier than the dimeric 8.6 complex, suggesting that the tetravalent 8.6 does not induce the formation of TLR3 assemblies larger than a dimer.”

Secondly, the activated form of TLR3 bound to the binders (the multimerized binder and coiled-coil fused mini-binder) is similar to TLR3-dsRNA structure? Ideally, authors determine the 3D structure by CryoEM, but 2D class imaging by CryoEM should be shown.

We thank the reviewer for this comment. We attempted unsuccessfully to resolve clear images of multimerized minibinder/TLR3 complexes via 2D class imaging. We believe that this is because both the multivalent and dimeric constructs have long, flexible GS linkers that do not drive the formation of highly ordered TLR3 oligomers.

Thirdly, the repeat number of GS linker (i.e. length of GS linker) affect the activity?

We thank the reviewer for raising this interesting question. As shown in a new Extended Data Fig. 10b, we evaluated different lengths of GS linkers in the multimerized binders, including 12, 16, and 40 residues. Our data indicate that variations in the linker length did not significantly affect TLR3 activation, as all constructs demonstrated comparable levels of NF- κ B signaling in TLR3-expressing cells. We have added the following text to the Results section to describe these data:

“Varying GS linker lengths (12, 16, and 40 residues) and minibinder valencies (2, 4, and 6 copies of minibinders) did not significantly impact TLR3 activation, as all constructs exhibited comparable levels of NF- κ B signaling in TLR3-expressing cells (**Extended Data Fig. 10b**).”

2) As the authors addressed, the mini-binders activated NF- κ B via TLR3 in a cell line that expresses TLR3 on the cell surface. TLR3 is localized in endolysosome, therefore mini-binders should bind to TLR3 at an acidic condition. Binding affinity of the mini-binders to TLR3 should be shown at an acidic condition.

This is a great point we were curious about as well, although we note that the TLR3hi cell line was previously shown to express TLR3 on the cell surface (Leonard et al., *PNAS* 2008). We tested binding at pH 5.5, 6.5, and 7.4 in a new Extended Data Fig. 10d. As previously reported, poly(I:C) showed pH-dependent binding to TLR3. By contrast, we observed no significant changes in the binding of minibinders 7.1 or 8.6 to TLR3 as a function of pH. We have added the following text to the Results section to describe these data and additional data in response to a request from Reviewer 2:

“Although the endosome acidification inhibitor chloroquine had no effect on minibinder-induced signaling (Extended Data Fig. 10c), binding of the minibinders to TLR3 was consistent from pH 5.5–7.4 (in contrast to poly(I:C); Extended Data Fig. 10d), leaving open the possibility that our constructs activate TLR3 either at the cell surface or in the endosome.”

3) The mini-binders do not inhibit dsRNA binding to TLR3?

We thank the reviewer for this important question. As shown in new data presented in Extended Data Fig. 10e,f, we used competitive ELISA to assess the potential of the minibinders to inhibit dsRNA binding. Our results indicate that minibinder 8.6 does not inhibit dsRNA binding to TLR3 in either monovalent or tetravalent form. By contrast, minibinder 7.1, especially in its tetravalent form, does show an inhibitory effect. Accompanying structural models depict how minibinder 7.1 overlaps partially with bound dsRNA, explaining this competitive inhibition. We have added the following text to the Results section to describe these data:

“Finally, although minibinder 7.1 inhibited dsRNA binding to TLR3, especially in its tetravalent form, neither monovalent nor tetravalent minibinder 8.6 inhibited dsRNA binding (Extended Data Fig. 10e). These observations were consistent with our structures of the minibinder-TLR3 complexes, which showed that only minibinder 7.1 would clash with bound dsRNA (Extended Data Fig. 10f).”

Reviewer #2:

The purpose of the study is to synthesise and test peptide agonists and antagonists of the innate immune receptor TLR3. This is a challenging objective because TLR3 is naturally activated by dsRNA localised to endosomes and involves dimerization and multimerization of the receptor. Peptides will not usually be efficiently taken up by the endosomal pathway so the study relies on the use of a HEK293 cell line in which TLR3 can signal from the cell surface.

The TLR3 ectodomain is a challenging target as it is highly glycosylated so the authors identified two sites on the concave surface of the solenoid relatively free of glycans. Initial in silico docking analysis identified candidate peptides and these were then screened by yeast surface display yielding 11 hits. Further affinity maturation yielded two ‘minbinders’ with nM affinities for site A (7.1 and 8.6) and subsequent structural studies revealed similar α -helical bundles bound at site A. This peptide design is impressive and a proof of principle for targeting difficult protein interfaces.

We thank the reviewer for their positive assessment of our work.

The next step was to test the ability of the minibinders to affect TLR3 signalling. To do this they made a dimeric and tetrameric forms of the binders and tested for activation in HEK293 (TLR3 high) cells. They chose to use an NF κ B reporter linked to GFP as a readout and these results are presented in Figure 4. I do not understand why they have not use more usual and more quantifiable outputs such as luciferase or SEAP.

We thank the reviewer for their comments. The TLR3hi cell line with an integrated GFP reporter was originally reported in Leonard et al., *PNAS* 2008, and we gratefully used this existing resource for our assays. We quantified TLR3hi activation by measuring the Mean Fluorescence Intensity of GFP as previously reported.

In Figure 4d what is BL-1A and in what way is this a histogram?

We apologize for the lack of clarity. “BL-1A” denotes the channel used during flow cytometry to monitor GFP fluorescence upon excitation by the 488 nm (blue) laser. We have updated Figure 4b to include a schematic of the flow cytometry setup used to clarify this for the reader. The histograms show the number of events observed at each fluorescence intensity value, a common representation used for flow cytometry data. We have also added a fourth panel to our flow gating schematic, now Extended Data Fig. 10a, so that the reader can see representative raw data.

The concentration of minibinders needed to produce a response is very high (300µg/ml). I find Fig4e, which is key to the whole study, difficult to understand. The symbols used for the different experiments are unclear – why does 7KO tetramer still signal?

We appreciate the reviewer’s observations regarding Figure 4e and the concentrations of minibinders used. We agree that the effective concentrations required to observe a response are relatively high (300 µg/mL), which likely reflects the need for optimal spatial clustering of TLR3 for activation. Designing minibinder multimers that achieve geometrically optimal receptor clustering will be the subject of future work.

Regarding the symbols used in Figure 4e, we have updated the figure layout to provide clearer identification of each experiment and the corresponding minibinder constructs.

The minor amount of residual signaling from the tetravalent 7 KO 16GS likely derives from multimerization providing an avidity boost to a very weak interaction between TLR3 and the knockout mutant. Nevertheless, we note that the observed signaling from the tetravalent 7 KO is lower than the signaling from the intact monovalent minibinder.

Why are there two 8.6 dimers one signalling and the other not (purple lined squares and purple filled squares)?

We thank the reviewer for identifying this error and have updated the figure legend accordingly. The purple lined squares are now labeled as “8 KO dimer”. We appreciate the reviewer’s attention to detail in catching this mistake.

What is the yellow filled triangle at the left half way up the y-axis?

The yellow triangle represents the monovalent 7.1 minibinder. To improve clarity, we have added a “monovalent” schematic to Fig. 4a to illustrate this construct. Additionally, we have separated

the two graphs in Fig. 4e into four individual graphs to enhance legibility and make the data easier to interpret.

In my opinion these experiments should be repeated with a luciferase reporter in a more relevant cell line eg THP1-Dual-hTLR3 cells. They should also be supplemented with TRIF cytokine assays like IFN and RANTES.

We thank the reviewer for this suggestion. Although we agree that THP1-Dual-hTLR3 cells may be a more biologically relevant model, we have some concern regarding the presence of TLR4 in these cells. Because our minibinders are produced in bacteria, low levels of residual endotoxin could be present in our preparations and drive confounding TLR4 signaling. This is one of the reasons we selected the TLR3hi cell line for our studies.

The cell surface origin of the signal should be validated with an endocytosis blocker. Perhaps also a control using ordinary HEK293s should be added – in our experience these do signal in response to polyIC but at a lower level

We thank the reviewer for these suggestions. We compared the activity of our minibinders in the presence and absence of chloroquine, an endosome acidification inhibitor. The presence of chloroquine decreased poly(I:C) signaling, but not tetravalent 7.1 or 8.6 minibinder signaling, consistent with activation of cell-surface TLR3 as described in Leonard et al., *PNAS* 2008. However, the pH-independent binding of 7.1 and 8.6 to TLR3 leaves open the possibility that they do activate TLR3 in the endosome. We have included these results in a new Extended Data Fig. 10c,d and described them with the following new text in the Results section:

“Although the endosome acidification inhibitor chloroquine had no effect on minibinder-induced signaling (Extended Data Fig. 10c), binding of the minibinders to TLR3 was consistent from pH 5.5–7.4 (in contrast to poly(I:C); Extended Data Fig. 10d), leaving open the possibility that our constructs activate TLR3 either at the cell surface or in the endosome.”

In conclusion the development of peptides agonists targeted at the TLR3 is an important result but the signalling properties of the helical peptides needs further characterization.

We appreciate the constructive feedback and trust that the functional properties of our minibinders are now clearer due to the additional experiments we have performed in response to Reviewers 1 and 2, along with the clarifications made to Fig. 4 in response to Reviewer 2's suggestions.

Reviewer #3:

This interesting and very well written manuscript presents the de novo design of TLR3 binding miniproteins, and the assembly of these binders into multivalent constructs with agonist activity. However, no novel protein design or engineering advances are reported. Therefore, the novelty seems to come primarily from the application.

Unfortunately, the functional characterization of the engineered constructs to act as an adjuvant does not go further than the initial proof-of-concept for TLR3 signaling using an artificial reporter cell line.

Indeed, the minibinder was not evaluated for activity as an adjuvant, presumably because a new construct would need to be engineered that binds to TLR3 from a model organism like a mouse or rat?

The reviewer is correct—the minibinders are not cross-reactive to mouse TLR3. As highlighted in Extended Data Fig. 8, there are a few differences between human and mouse TLR3 that prevent cross-reactivity.

Nor was the binding affinity, valency, or linker distance systematically varied and evaluated in order to tune the agonist activity to achieve the desired effect as an adjuvant.

We thank the reviewer for this suggestion. As shown in the new Extended Data Fig. 10b, we evaluated different GS linker lengths in the multimerized binders, including 12, 16, and 40 residues, and tested configurations with 2, 4, and 6 copies of the minibinders. Our data indicate that variations in linker length and the number of minibinder copies did not significantly impact TLR3 activation, as all constructs exhibited comparable levels of NF- κ B signaling in TLR3-expressing cells. Furthermore, Extended Data Fig. 9b shows that dimeric and tetravalent 8.6 in complex with TLR3 have similar SEC profiles, suggesting that the multivalent minibinders primarily induce TLR3 dimerization rather than higher-order assemblies. Future work will aim to engineer constructs capable of geometrically precise multimerization of many copies of TLR3 to enhance signaling.

Nonetheless, the protein design work is well performed and the biochemical and structural characterization of the designs is thorough. Very few academic labs in the world are able to engineer a minibinder in the way that the authors describe, so the application of this technology remains exciting.

We thank the reviewer for their positive assessment of our work.

The authors position this as a first step with functional work to come in the future. However, without compelling functional data, this story should not focus so thoroughly on the use of these

constructs as an adjuvant, and should focus instead more on the protein design, biochemistry, and structural biology aspects, as that is where the strength of the data lies. Pending this repositioning of the story to align the narrative with the data reported, I highly recommend this manuscript for publication in Nature Communications.

We thank the reviewer for this comment. We have edited the revised manuscript to address this point. We kept the description of the rationale of protein-based adjuvants in the Introduction, as this is a necessary motivation for the manuscript. However, we significantly reduced the forward-looking comments on the potential of protein-based adjuvants in the Discussion section since, as the reviewer points out, we do not yet have data to substantiate those comments. To this end, we eliminated the first paragraph of the discussion, simply subsuming one sentence into the paragraph discussing our cell-based assays demonstrating activation of TLR3.

Additionally, I have the following comments -

On page 3, the sentence “We first used RifGen 48 to dock disembodied amino acid side chains against Sites A and B, and in parallel docked 21,402 pre-existing polyvaline scaffolds (i.e., backbones) against the same sites using PatchDock.” The terms “scaffold” and “backbone” are both protein design jargon and using one does not necessarily help the uninitiated reader understand the other. I think it might be helpful to explain these terms in plain language, i.e. a “scaffold” or “backbone” is a protein main chain or tertiary structure.

We thank the reviewer for this suggestion. We have updated the text to say “...and in parallel docked 21,402 pre-existing protein backbones comprising a diverse array of 3-helix bundles against the same sites”.

Authors mention that “TLR3 is a challenging target for several reasons” but don’t describe any new computational innovations that were required to get binders for this target. If previously developed workflows could still be employed and binders were found at a hit rate that is on par with previous studies, is it fair to say that this is a challenging target?

The reviewer raises a fair point. We have revised the first paragraph of the Results section to read:

“In *de novo* minibinder design, target site selection significantly influences the success rate of obtaining functional binders. TLR3, like other TLRs, is highly glycosylated, particularly on the surfaces of the molecule not involved in dsRNA-mediated dimerization. Moreover, the carbohydrate-free surface of the molecule features many polar residues with few solvent-exposed hydrophobic patches. The combination of these two features in TLR3 made target site selection difficult, as existing minibinder design methods are most successful when targeting hydrophobic surface patches. Furthermore, since our goal was to design minibinder agonists that can drive TLR3 signaling by

multimerizing the receptor, we required a target site that would allow oligomeric versions of the minibinder to simultaneously engage multiple TLR3 ectodomains.”

For the cryo-EM structure determination, I'm a bit confused as to why mixing minibinder with the TLR3 ectodomain failed to produce sufficient occupancy, but co-expression was. It would be helpful to include a bit of discussion to explain what you think is happening here.

We thank the reviewer for this question. Simply mixing TLR3 ectodomain and minibinder led to only ~10-20% occupancy of the minibinder. In previous studies, TLR4 was co-expressed and co-purified with MD-2 for structural characterization, and that strategy proved successful in improving the occupancy of our minibinders. Potential explanations may include that binding is favored by the high concentrations of both TLR3 and the minibinders inside the ER lumen during secretion, or that the glycan at N413 hinders binding unless it occurs during folding (although this would be inconsistent with our BLI data). However, we prefer not to include such speculations in the manuscript. We have added text to the following passage in the Results section to describe our empirical approach to improving occupancy based on previous structural studies of TLRs:

“Although a minority of the 2D class averages had density potentially corresponding to the minibinder at target Site A, overall the data suggested low minibinder occupancy. Following a strategy previously established for structural characterization of TLR4/MD-2 complexes, we co-expressed the TLR3 ectodomain and each minibinder in HEK293F cells and co-purified the complexes using affinity chromatography followed by SEC. This improved occupancy, and after optimizing grid preparation and data collection to overcome a clear preferred orientation, we determined cryo-EM structures of minibinders 7.7 and 8.6 in complex with TLR3, each at 2.9 Å resolution.”

Fusing multiple minibinders together is a great strategy, but the nomenclature used here is confusing because it conflicts with the broader protein biochemistry literature. “Tetramer” is a term used to describe 4 separate protein chains that bind each other to form a quaternary structure complex. This term is not commonly used to describe genetic fusions of a protein into a single construct, and I worry the way that it is used here will be confusing to readers. For example, an IgG heavy chain consists of 4 separate Ig domains in a single protein construct, but this is never referred to as a tetramer of Ig domains in the literature; it's one protein chain. Perhaps “multivalent”, “tetra-valent” and “bi-valent” are better terms to use here? Whatever the authors choose, the term should not have a different meaning that is already well established in the literature.

The reviewer makes a fair point. We have updated “tetramer” to “tetravalent” throughout the manuscript and figures. Similarly, we have called additional valencies of the binders “monovalent,” “bivalent,” and “hexavalent” in Figure 4 and throughout the text. We have left the antiparallel dimer as “dimer”.

It looks like the BLI for the constructs named ‘8.6 dimer’ and ‘8.6 tetramer’ have a slower koff, which is what you would expect from getting an avidity boost. It might be helpful to include a sentence describing as much, and perhaps to mention the difference in koff that you measured for the monovalent and multivalent constructs. Also, what is the final KD that you measured for the multi-valent constructs?

We thank the reviewer for this suggestion. We have now included BLI data for all multivalent constructs in Extended Data Fig. 9a. In this assay, we loaded hTLR3-Fc onto the tips and associated the multivalent constructs, which differs from our monovalent BLI experiments where monomeric TLR3 was loaded. Due to the multivalent nature of these interactions and potential avidity effects, calculating a precise K_D value is challenging. We added the following text to the Results section to describe these data:

“We expressed and purified the multivalent 7.1 and 8.6 minibinders as well as the 8.6 antiparallel dimer and confirmed that they bind to TLR3 by BLI (Extended Data Fig. 9a). The multivalent binders appeared to have slower dissociation rates than their monovalent counterparts as expected for avid interactions, although this prevented the calculation of dissociation constants. As negative controls, we also purified “interface knockout” versions of 7.1 and 8.6 that incorporate mutations shown to decrease TLR3 binding in our earlier SSM data (Supplementary Table 1).”

The minibinder 7.7 needs to partially unfold to bind. How stable is this protein to serum proteases? Would you expect it to survive in vivo long enough to be an effective adjuvant?

We thank the reviewer for this question. This would certainly be an important consideration for *in vivo* studies, and one that we look forward to exploring in future work.

Would a strong immune reaction against these minibinders neutralize their activity and prevent them from functioning as adjuvants?

This is another interesting question and one that we have considered, as it also comes up when discussing antibody responses against the self-assembling protein scaffolds that underlie our lab’s nanoparticle immunogens (see, for example, Kraft et al., *Cell Rep. Med.* 2022). In that context, our work and others’ to date indicates that anti-scaffold responses do not appear to deleteriously affect immune responses against displayed antigens. However, as the reviewer suggests, antibodies against a protein-based adjuvant certainly could neutralize its activity. We look forward to evaluating this question in future studies.

The authors never define what “SC-50” means, e.g. Figure 2D.

We thank the reviewer for this pointing out this oversight. We have updated our description in the Figure caption as follows:

“Site saturation mutagenesis heat maps of interface residues. The originally designed amino acid at each position is provided at the bottom and in the white square. Red indicates affinity improvement and blue indicates affinity reduction. *SC-50 is the midpoint concentration in the binding transition, a proxy for the K_D , and is described in detail in ref. 42.*”

Furthermore, the methods section does not contain any information for how the SSM library was sorted, how the data were analyzed to produce the heatmaps, or what the measurements represent.

We thank the reviewer for pointing out this omission. We have added the following text to the Materials and Methods section:

“For SSM libraries, two rounds of no-avidity sorts were performed and in the third round of screening the libraries were titrated with a series of decreasing concentrations of hTLR3. After each sort, cells were collected for deep sequence analysis. For the combinatorial libraries, five rounds of no-avidity sorts were performed with decreasing concentrations of hTLR3. Only the top 0.1% of the binding population was collected after each sort. These populations were plated on C-trp-ura plates and the sequences of individual clones were determined by Sanger sequencing.”

“Deep sequence analysis

Collected sorts for the naive and SSM libraries were sequenced using Illumina NextSeq sequencing. The PEAR program⁸¹ was used to assemble the fastq files as previously described⁴². For the SSM libraries, the apparent SC-50 was estimated using the fitting procedure described in ref. 42. SC-50 is the midpoint concentration in the binding transition, a proxy for K_D . Heatmaps were generated in Jupyter notebooks based on the scripts in ref. 42.”

In the Combinatorial library preparation section what does it mean that “ultramers were stitched together and cloned into yeast” ? Does this mean that DNA was assembled enzymatically in vitro prior to yeast transformation? If so, how?

We thank the reviewer for this question. We have added the following text to the Methods section to describe this more clearly for the reader:

“We identified mutations that enhanced affinity from SSM heatmaps. We then used SwiftLib to create libraries containing these mutations with degenerate codons⁸³. We

ordered two overlapping Ultramers (long ssDNA oligos) for each design that contained the selected degenerate codons (Integrated DNA Technologies), assembled them by PCR, and cloned into yeast. All libraries were combined before yeast surface display.”

How were the yeast transformed, i.e. chemical competence or electroporation? What fold-coverage was obtained for the library?

We thank the reviewer for pointing out this omission. We have specified in the revised Methods that “*Minibinder libraries were transformed into yeast via electroporation.*” We did not measure fold-coverage.

Reviewer #1:

The authors have addressed my previous concerns. I recommend the publication of this manuscript in Nature communications.

We thank the reviewer for their positive assessment of our work.

Reviewer #2:

On the whole the authors have responded adequately to my comments, especially they have tidied up Figure 4 and it is now more comprehensible.

We thank the reviewer for their positive remarks.

However they have side-stepped two of my points: firstly they have done some experiments with chloroquine to satisfy the concern about pH dependence (lack of) and this looks convincing. It should be noted that chloroquine has lots of other off-target effects and the more specific bafilomycin should have been used. They refer to this same data as a response to my point about endocytosis but chloroquine is not an endocytosis inhibitor, they should use dynasore.

We thank the reviewer for these suggestions. We have now tested the effects of bafilomycin A1, Pitstop[®] 2 (an inhibitor of clathrin-mediated endocytosis), and dynasore on TLR3 signaling. These data are included in Fig. 1 below.

Bafilomycin completely abrogates signaling from Poly(I:C). This is expected, as Poly(I:C) cannot bind to TLR3 at neutral pH. Notably, the presence of bafilomycin slightly increased signal from the tetravalent binders. This result initially surprised us, but has been seen in the literature previously: Leonard et al. showed that bafilomycin increased the signal of an agonistic anti-TLR3 antibody (Figure 4D, Leonard et al., *PNAS* 2008).

Pitstop[®] 2 decreased Poly(I:C) signaling, but did not result in a complete reduction. Similarly, Pitstop[®] 2 slightly decreased the signaling of the tetravalent binders. This indicates some signal may be derived from endosomal TLR3, but the majority is from cell surface TLR3. Dynasore showed minimal effects on Poly(I:C) and tetravalent binder signaling.

As these data do not allow definitive quantification of the contributions of cell-surface and endosomal TLR3 to signaling, we have not included them in the revised manuscript.

Fig. 1: Effects of various inhibitors on TLR3 signaling. Cells were preincubated with inhibitors 1 hour prior to stimulation. For untreated cells, PBS was added 1 hour prior to stimulation. GFP levels were measured with flow cytometry. Mean fluorescence intensity (MFI) values from assay duplicates are plotted.

Secondly, they claim that validation THP-1 macrophages would be confounded by LPS contamination. I do not accept that - it would be straightforward to inhibit TLR4 with the very specific TAK-242 as a control for LPS.

We thank the reviewer for this suggestion. We have tested our constructs in THP1-Dual hTLR3 cells pretreated with 10 µM of TAK-242. We measured IRF and NF-κB responses. These data are included in Fig. 2 below.

TAK-242 effectively inhibits IRF and NF-κB signaling at low concentrations of LPS, as expected (Fig. 2a). However, above 100 µg/mL, the inhibitor begins to lose its effectiveness. TAK-242 had minimal effects on Poly(I:C)-induced signaling, as expected (Fig. 2b).

Tetraivalent and monovalent minibinder 8.6 both readily signal in this assay (Fig. 2c). However, our constructs were prepared in *E. coli* and tested at endotoxin levels of >500 EU/mL. While

there is a slight decrease in signal in the presence of TAK-242, the endotoxin levels in these proteins are high. As shown in Fig. 2a, the inhibitor begins to lose effectiveness at concentrations above 100 $\mu\text{g}/\text{mL}$ of LPS. We suspect that treatment with our minibinders exceeds this threshold, compromising the ability of TAK-242 to prevent TLR4 signaling and confounding isolation of TLR3-induced signaling. For this reason, we believe that it is more accurate and appropriate to measure TLR3 signaling in TLR4 KO cell lines, and have not included the THP-1 data in the revised manuscript.

Fig. 2: Effect of TAK-242 on cell signaling. THP1-Dual hTLR3 cells were preincubated with 10 μM of TAK-242 (+) or PBS (-) 1 hour prior to stimulation. After stimulation with various ligands (a, LPS. b, Poly(I:C) c, TLR3 minibinders), cells were incubated for 24 hours. *Top*, The NF- κB response was assessed using SEAP and QuantiBlue (OD 620 nm). *Bottom*, The IRF response was assessed using Lucia luciferase (QuantiLuc). Data is shown as fold response over non-induced cells.

Reviewer #3:

The authors have address all my questions and comments, and I now recommend this manuscript for publication in Nature Communications.

We thank the reviewer for their positive assessment of our work.